# Density-Aware Prototypical Network for Few-Shot Relation Classification

**Jianfeng Wu[1]    Mengting Hu[2*]   Yike Wu[3]    Bingzhe Wu[4]**
**Yalan Xie[1]    Mingming Liu[2]    Renhong Cheng[1]**

[1] College of Computer Science, Nankai University, [2] College of Software, Nankai University
[3] School of Journalism and Communication, Nankai University, [4] Tencent AI Lab
`wjf@mail.nankai.edu.cn`, `mthu@nankai.edu.cn`

## Abstract

In recent years, few-shot relation classification has evoked many research interests. Yet a more challenging problem, i.e. *none-of-the-above* (NOTA), is under-explored. Existing works mainly regard NOTA as an extra class and treat it the same as known relations. However, such a solution ignores the overall instance distribution, where NOTA instances are actually outliers and distributed unnaturally compared with known ones. In this paper, we propose a density-aware prototypical network (D-Proto) to treat various instances distinctly. Specifically, we design unique training objectives to separate known instances and isolate NOTA instances, respectively. This produces an ideal instance distribution, where known instances are dense yet NOTAs have a small density. Moreover, we propose a NOTA detection module to further enlarge the density of known samples, and discriminate NOTA and known samples accurately. Experimental results demonstrate that the proposed method outperforms strong baselines with robustness towards various NOTA rates.[1]

## 1   Introduction

Relation classification is a fundamental task in natural language processing field, aiming to recognize the relation between two entities in a given sentence. Recently, many works formulate this task in the few-shot learning scenario (Dong et al., 2020; Qu et al., 2020; Han et al., 2021; Brody et al., 2021). They focus on obtaining fast adaptation with a few samples by learning multiple episodes. Usually, each episode is composed of a support set and a query set, where the support set has $N$-way $K$-shot instances. As an example shown in Table 1, the support set provides two known relations, i.e. *"contains"* and *"founded by"*. The first query instance has *"founded by"* relation, which can be

---

*Mengting Hu is the corresponding author.
[1]The code is released on GitHub at `https://github.com/Pisces-29/DProto`.

| Support Set | |
|---|---|
| (1) contains | *California* is a state in *the United States*. |
| (2) founded by | *Steve Jobs* was one of the founders of *Apple*. |

| Query Set | |
|---|---|
| (2) | *Bill Gates* founded *Microsoft* in 1975 |
| NOTA | *Einstein* was born in *1879*. |

Table 1: An example for a 2-way 1-shot episode with 50% NOTA rate, where 2 indicates the number of classes and each class has one instance. The head entity is marked in red, and the tail entity is marked in blue.

recognized by reference to this support set. But for the second query instance, the relation between *"Einstein"* and *"1879"* is *"born in"*. It cannot find a reference from the support set.

Gao et al. (2019b) define such relation as *none-of-the-above* (NOTA) and propose a new task setting. Its challenge relies on the accurate detection of both the known instances and NOTA ones. Previous works extend $N$-way to $(N+1)$-way, regarding NOTA relation as an extra class. Then the problem is further solved by classification (Gao et al., 2019b), or computing distance with pre-defined learnable vectors (Sabo et al., 2021), or multiple choice (Liu et al., 2022). Despite their popularity and superiority, they only focus on detecting NOTA effectively rather than refining the distribution of NOTA instances, leaving the overall distribution of all instances ignored. However, NOTA is not a simple $(N+1)$-th class, which tends to be outliers for the known $N$ relations. As an illustration, MNAV may lead some NOTA instances to cluster around a particular vector. However, the clustering of these instances from different categories together is unnatural. Therefore, motivated by this, we study from the view of instance distribution.

In this paper, our initial research question is *what*

*instances distribution is ideal.* For known samples, intra-class needs to be well clustered and inter-class is clearly separated. And NOTA samples, without gathering, should be distributed away from all known instances. In light of these properties, we propose a density-aware prototypical network (**D-Proto**) that takes into account the density of different types of instances. Specifically, to guarantee the ideal instance distribution of known instances, we introduce neighborhood component analysis (NCA) (Laenen and Bertinetto, 2021), which involves all sample pairs out of the support and query sets. For NOTA instances, we propose a NOTA loss, which aims at pushing them away from all non-NOTA ones. By making NOTA instances isolated, both of them become more distinguishable in the embedding space.

Moreover, the ideal distribution will lead to distinct densities for NOTA and known samples. That is to say, known instances are distributed densely since the intra-class is clustered together. NOTA's density is small as it is isolated from others. Therefore, we propose a NOTA detection module by fully enhancing the density properties. Concretely, to promote the density of known instances, we design intra-class prototype enhancement. By adding prototypes and degenerative prototypes, their densities could be further enlarged. Based on this, we propose a prototype enhanced local outlier factor (PLOF) to calculate the outlier extent of an instance through distances with its neighbors. Finally, we propose a base score that uses the extremal distance ratio between query samples and $N$ prototypes. By fusing PLOF and base score, the final NOTA score is obtained to accurately discriminate between NOTA instances and known ones.

In summary, the main contributions of our work are as follows:

- We propose D-Proto to address the more challenging few-shot relation classification task. To the best of our knowledge, this work is the first to model the overall instance distribution in an episode.

- We introduce the NCA objective and propose a NOTA loss to achieve ideal instance distribution. Based on its ideality, we further propose a prototype enhanced local outlier factor (PLOF) and a base score to fully leverage the density properties.

- Extensive experimental results demonstrated

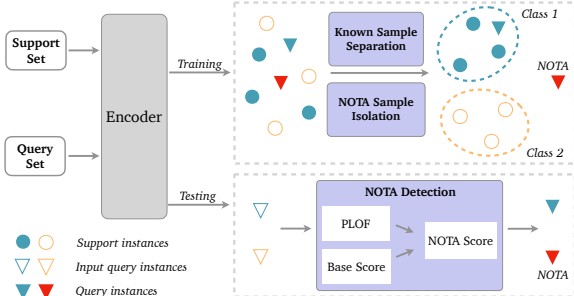

Figure 1: The framework overview of D-Proto. In this example, the support set is 2-way 3-shot. During training, two different query samples are for class 1 and NOTA, respectively. During testing, queries are first predicted via distances with prototypes and further revised with a NOTA detection module.

the effectiveness and robustness of our approach on both fixed and random episode sampling with various NOTA rates.

## 2 Methodology

### 2.1 Formulation and Method Overview

Given the training set $\mathcal{D}_{train}$, we sample multiple episodes, where each of them is composed of $\{\mathcal{S}, \mathcal{Q}, \mathcal{R}\}$. The relation set $\mathcal{R} = \{r^1, r^2, ..., r^N\}$ mentions $N$ relations. The support set $\mathcal{S}$ follows $N$-way $K$-shot setting, which contains $N$ relations, and each relation $r^i$ has $K$ instances.

$$\mathcal{S} = \{(s_j^i, r_j^i)\}, 1 \le i \le N, 1 \le j \le K$$

where $s_j^i$ denotes the $j$-th sample belonging to the relation $r^i$. The query set $\mathcal{Q}$ is composed of two kinds of instances: known instances, which belongs to the relation set $\mathcal{R}$ of the support set; NOTA instances, without references from the support set.

$$\mathcal{Q} = \{(q_1, r_1), ..., (q_n, r_n), (\overline{q}_1, \overline{r}), ..., (\overline{q}_m, \overline{r})\}$$

where $\{(q_j, r_j)\}_{j=1}^n$ are the known instances and $\{(\overline{q}_j, \overline{r})\}_{j=1}^m$ are the NOTA instances. Here $\overline{r}$ denotes the NOTA relation. The query set $\mathcal{Q}$ is predicted with the help of support set $\mathcal{S}$. Our goal is to recognize all samples in $\mathcal{Q}$ accurately.

The proposed approach is depicted in Figure 1. A sample is encoded with BERT $E(\cdot)$ (Devlin et al., 2019), yielding the vector representation of [CLS]. In the embedding space, we design two training objectives: 1) **known sample separation** $\mathcal{L}_{nca}$ aims to increase the separability of known samples by reducing intra-class distances and enlarging inter-class distances; 2) **NOTA sample isolation** $\mathcal{L}_{nota}$

push NOTA samples to be distinct from known ones, making them more easily identifiable as outliers. Ultimately, our model is optimized by the combination of two training objectives.

$$\mathcal{L} = \mathcal{L}_{nca} - \alpha\mathcal{L}_{nota} \tag{1}$$

where $\alpha$ is a hyper-parameter to balance the contribution of NOTA loss.

In the testing phase, we propose **NOTA detection** module, fusing PLOF and base score into NOTA score. Each sample's NOTA score then reveals its outlier extent. NOTA samples will be easily discriminated from known samples. Next, we will introduce each module in detail.

## 2.2 Known Sample Separation

Although considering the overall instance distribution is intuitive, its ideal is not easy to attain in the few-shot scenario. Due to the limitation of samples, a sample and its neighbors are all important for quantifying density. Therefore, we introduce neighborhood component analysis (NCA) loss (Laenen and Bertinetto, 2021). It considers the distances between all sample pairs from both the support set and the query set. In other words, $K$ samples of some relation are pulled together interactively and away from all other $K \times (N-1)$ samples.

Assume all instances in an episode are $\mathcal{B} = \{(x_i, r_i) | (x_i, r_i) \in \{\mathcal{S} \cup \mathcal{Q}\}, 1 \leq i \leq b\}$, then all the known instances are defined as $\mathcal{B}_k = \{(x_i, r_i) | (x_i, r_i) \in \mathcal{B}, r_i \neq \overline{r}\}$, where $x_i$ is a instance, $r_i$ is its relation, $\overline{r}$ represents the NOTA relation. NCA loss is formulated as below.

$$\mathcal{L}_{nca} = \frac{-1}{|\mathcal{B}_k|} \sum_{\substack{i \in 1,...b \\ r_i \neq \overline{r}}} \log$$
$$\left( \frac{\sum_{\substack{j \in 1,..,b \\ j \neq i \\ r_i = r_j}} \exp(-d^2(\boldsymbol{x}_i, \boldsymbol{x}_j))}{\sum_{\substack{k \in 1,...,b \\ k \neq i \\ r_k \neq \overline{r}}} \exp(-d^2(\boldsymbol{x}_i, \boldsymbol{x}_k))} \right) \tag{2}$$

where $\boldsymbol{x}_i = E(x_i)$ is the vector representation for the instance $x_i$. $d(\cdot, \cdot)$ represents the Euclidean distance. $|\mathcal{B}_k|$ indicates the number of samples in $\mathcal{B}_k$. With $\mathcal{L}_{nca}$, the intra-class is clustered together and the inter-class is away from each other, making a good distribution for the density of known samples.

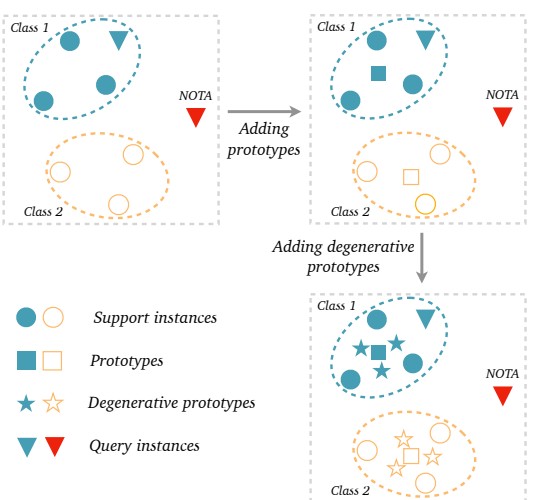

Figure 2: Illustration of expanding instance sets for each relation in PLOF. We specifically augment with prototypes and degenerative prototypes.

## 2.3 NOTA Sample Isolation

Since NOTA samples are outliers for $N$ known relation clusters, they should be isolated from known ones. Meanwhile, if a query set has multiple NOTAs, they might belong to different relations like the second query instance of Table 1. Aggregating NOTA samples into the $(N+1)$-th class may cause a negative impact on distinguishing relations. Therefore, we propose NOTA loss to push each NOTA instance away from *all the known samples*. The NOTA instances are denoted as $\mathcal{B}_{nota} = \{(x_i, r_i) | (x_i, r_i) \in \mathcal{B}, r_i = \overline{r}\}$. $\mathcal{L}_{nota}$ is defined as below.

$$\mathcal{L}_{nota} = \frac{-1}{|\mathcal{B}_{nota}|} \sum_{\substack{i \in 1,...,b \\ r_i = \overline{r}}} \log$$
$$\left( \sum_{\substack{j \in 1,...,b \\ r_j \neq \overline{r}}} \exp(-d^2(\boldsymbol{x}_i, \boldsymbol{x}_j)) \right) \tag{3}$$

In this way, each NOTA sample is isolated from all others, which is beneficial for detecting known relations and NOTA via density.

## 2.4 NOTA Detection

To further quantify densities accurately for few-shot samples, we promote the testing phase with PLOF and base score. They are aggregated into the final NOTA score, which indicates an instance's outlier degree through its nearest instances and prototypes. Next, we will introduce them individually.

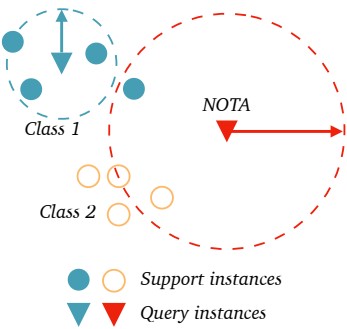

Figure 3: A 3-neighborhood example. The radius of each circle represents 3-distance.

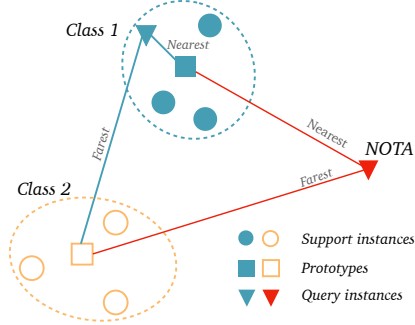

Figure 4: An example of a 2-way 3-shot scenario. For a known query, its base score is much smaller than that of a NOTA query.

**Prototype Enhanced Local Outlier Factor (PLOF)** Different from the naive prototypical network (Snell et al., 2017), we compute the sample pair distances as Eq. (2). However, during inference, we argue that the prototypes can be leveraged to enlarge the density. Therefore, as depicted in Figure 2, we first calculate the prototype for each relation as below and expand the relation's vector set with its prototype.

$$c^{r^i} = \frac{1}{K} \sum_{j=1}^{K} E(s_j^i) \quad (4)$$

To further boost the density and improve the identification of NOTA samples, we propose a special type of prototype, referred to as a "*degenerative prototype*", which is computed with a degenerative set of instances.

$$c_m^{r^i} = \frac{1}{K-1} \sum_{\substack{j=1,...,K \\ j \neq m}} E(s_j^i), 1 \leq m \leq K \quad (5)$$

Intuitively, each degenerative prototype $c_m^{r^i}$ is a vector obtained by averaging $(K-1)$ support embeddings belonging to its relation $r^i$ after removing a support instance $s_m^i$. As shown in Figure 2, the augmentation of prototypes and degenerative prototypes significantly increases the density of known instances and the outlier status of NOTAs, thus improving the detection.

Before calculating the PLOF of each instance, the original instance set $\mathcal{B}$ is expanded to $\mathcal{B}_{plof} = \mathcal{B} \cup \mathcal{P} \cup \mathcal{DP}$.

$$\mathcal{P} = \{c^{r^i}\}, 1 \leq i \leq N$$
$$\mathcal{DP} = \{c_m^{r^i}\}, 1 \leq i \leq N, 1 \leq m \leq K$$

where $\mathcal{P}$ is the prototype set and $\mathcal{DP}$ is the degenerative prototype set.

Then given a positive integer $k$ ($k < |\mathcal{B}_{plof}|$), the $k$-distance $d_k(q)$ of an vector $q$, is defined as that for an instance $o \in \mathcal{B}_{plof} \backslash \{q\}$, there are at least $(k-1)$ instances $v$ of which holds that $d(q, v) < d(q, o)$, where $v \in \mathcal{B}_{plof} \backslash \{q, o\}$. Here the symbol $\backslash$ indicates *except*. After finding such instance $o$, $d_k(q) = d(q, o)$. The Euclidean distance between $q$ and all the chosen instances are less than or equal to $d_k(q)$. As shown in Figure 3, dashed circles surround the chosen instances for each target $q$. These instances are called the $k$-neighborhood set of $q$, denoted as $N_k(q) = \{v \in \mathcal{B}_{plof} \backslash \{q\} | d(q, v) \leq d_k(q)\}$. Based on $k$-neighborhood, the local reachability distance $lrd$ is defined as follows:

$$lrd(q) = \frac{|N_k(q)|}{\sum_{o \in N_k(q)} \max\{d_k(o), d(q, o)\}} \quad (6)$$

Obviously, $lrd(q)$ is the inverse of the average reachability distance based on the $k$-neighborhood of $q$. Based on it, we further define PLOF as the average of the ratio of the local reachability density of $q$ and those of $q$'s $k$-neighborhood.

$$PLOF(q) = \frac{\sum_{v \in N_k(q)} \frac{lrd(v)}{lrd(q)}}{|N_k(q)|} \quad (7)$$

By considering the neighbors, PLOF better reflects the density. Meanwhile, the higher the PLOF value of $q$, the more probable that $q$ is an outlier (e.g. a NOTA sample).

**Base Score** Query instances are recognized with the help of support prototypes. We observe an interesting property on the distances between the query instances and support prototypes. As depicted in Figure 4, the non-NOTA query is close to its own prototype but far away from the other prototypes. Yet the phenomenon is different for the

NOTA query, which is far away from all prototypes. This occurs because the proposed NOTA loss $\mathcal{L}_{nota}$ pushes NOTA instances away from those known instances. Based upon this distance property, we propose a base score, which calculates the ratio of extremal distance to prototypes.

$$Score_{base}(\boldsymbol{q}) = \frac{\min\limits_{i=1,...,N} d^2(\boldsymbol{q}, \boldsymbol{c}^{r^i})}{\max\limits_{i=1,...,N} d^2(\boldsymbol{q}, \boldsymbol{c}^{r^i})} \quad (8)$$

This definition could also help to distinguish non-NOTA and NOTA samples. The base score of a known query is smaller, while it is larger for a NOTA query.

**NOTA Score**  Finally, the NOTA score is the combination of PLOF and base score.

$$Score_{nota}(\boldsymbol{q}) = PLOF(\boldsymbol{q}) \cdot Score_{base}(\boldsymbol{q}) \quad (9)$$

During inference, a query instance can be detected by comparing its NOTA score with a threshold $\tau$. Since the proposed NOTA score is well-designed, using this score can more effectively recognize instances. In the real application, the threshold can be estimated with k-fold validation. In our experiments, we choose a fixed threshold for simplification.

## 3  Experiments

### 3.1  Datasets

FewRel 1.0 (Han et al., 2018) is a relation classification dataset containing 100 relations extracted from Wikipedia. Gao et al. (2019b) update it to FewRel 2.0 and propose the NOTA detection challenge. Our experiments follow the splits used in official FewRel benchmarks, which split the dataset into 64 classes for training, 16 for validation, and 20 for testing.

### 3.2  Episode Sampling

In §2.1 we introduce the $N$-way $K$-shot setting. Each episode sampling support set follow the $N$-way $K$-shot setting. But there are two ways to sample a query set.

- **Fixed Sampling**  Gao et al. (2019b) propose fixing the NOTA rate of a query set to some specific value. For example, if a query set has 5 known queries and 5 NOTA ones, the NOTA rate of the query set would be calculated as $5/(5+5) \times 100\% = 50\%$. Then the total number of instances in that query set will be adjusted based on the NOTA rate.

- **Random Sampling**  Sabo et al. (2021) propose to fix the number of instances in the query set and assign a probability $p$ to each instance for sampling. If $p$ is less than a threshold $\tau_{na}$, the instance is sampled as a known query. Otherwise, it is sampled as a NOTA. As such, the value of $\tau_{na}$ determines the NOTA rate. This sampling strategy results in a dynamic NOTA rate for each episode. For example, some episodes may have a NOTA rate of 100%, while others may not have any NOTA instances, resulting in a more realistic simulation of real-world scenarios.

### 3.3  Baseline Methods

The following baseline methods are chosen for comparison. **O-Proto** (Tan et al., 2019) aims to solve few-shot out-of-domain detection. It is based on the prototypical network, which detects NOTA by cosine similarity. **BERT-Pair** (Gao et al., 2019b) adopts BERT to match a query instance and a support instance, which then yields a score indicating whether they share the same relation. **MNAV** (Sabo et al., 2021) is also based on the prototypical network. NOTA class is represented by some learnable vectors. A query instance is evaluated by its similarity with the NOTA vectors in the embedding space. **MCMN** (Liu et al., 2022) uses a pre-training and fine-tuning paradigm that converts all candidate relation descriptions into multiple-choice prompts.

### 3.4  Experimental Results

#### 3.4.1  Overall Results

The results of two sampling strategies are presented in Table 2. It can be observed that D-Proto outperforms strong baselines and achieves the best average results on both sampling strategies. Especially, D-Proto significantly outperforms MCMN by +2.75% on accuracy and +2.66% on F1 under random sampling. The results indicate the effectiveness of D-Proto, with great stability and robustness. As mentioned in §3.2, random sampling better reflects practical applications. It can be seen that under fixed sampling, the proposed D-Proto achieves competitive results. And under random sampling, the improvements are more significant. This implies that D-Proto is more robust in real-world situations.

Moreover, under random sampling, we see that the accuracy and F1 of MCMN decrease by 13.36% and 16.05%, respectively when the NOTA rate is

| | Random Sampling | | | | | | | |
|---|---|---|---|---|---|---|---|---|
| Methods | 15%NOTA | | 30%NOTA | | 50%NOTA | | Average | |
| | Acc | F1 | Acc | F1 | Acc | F1 | Acc | F1 |
| O-Proto (Tan et al., 2019) | 80.02 | 72.84 | 80.55 | 73.73 | 81.16 | 72.41 | 80.58 | 72.99 |
| BERT-Pair (Gao et al., 2019b) | 80.77 | 73.99 | 81.53 | 75.06 | 83.24 | 75.14 | 81.85 | 74.73 |
| MNAV (Sabo et al., 2021) | 85.02 | 78.81 | 83.67 | 77.78 | 81.59 | 73.94 | 83.42 | 76.84 |
| MCMN (Liu et al., 2022) | **88.24** | **83.54** | 82.50 | 76.39 | 74.88 | 67.49 | 81.87 | 75.80 |
| ChatGPT | 64.57 | 56.63 | 63.30 | 55.81 | 61.66 | 54.16 | 63.18 | 55.54 |
| D-Proto (Ours) | 85.76 | 79.79 | **84.80** | **79.27** | **83.30** | **76.32** | **84.62** | **78.46** |
| | Fixed Sampling | | | | | | | |
| Methods | 15%NOTA | | 30%NOTA | | 50%NOTA | | Average | |
| | Acc | F1 | Acc | F1 | Acc | F1 | Acc | F1 |
| O-Proto (Tan et al., 2019) | 81.38 | 80.45 | 81.42 | 81.04 | 81.38 | 79.39 | 81.38 | 80.29 |
| BERT-Pair (Gao et al., 2019b) | 82.83 | 81.98 | 83.76 | 83.00 | **85.49** | 82.62 | 84.02 | 82.53 |
| MNAV (Sabo et al., 2021) | 85.05 | 84.22 | 84.36 | 84.45 | 82.91 | 82.15 | 84.10 | 83.60 |
| MCMN (Liu et al., 2022) | **87.56** | 84.04 | 83.16 | 83.40 | 75.07 | 79.14 | 81.93 | 82.19 |
| ChatGPT | 63.59 | 56.26 | 61.83 | 54.46 | 58.41 | 51.14 | 61.28 | 53.95 |
| D-Proto (Ours) | 85.37 | **84.67** | **84.87** | **84.90** | 83.05 | **82.66** | **84.43** | **84.08** |

Table 2: Evaluation results of baseline methods and the proposed D-Proto, in terms of accuracy (%) and F1 (%), on the FewRel dataset. The setting is 5-way 5-shot.

set from 15% to 50%. On the contrary, D-Proto only sees decreases of 2.46% and 2.01%, respectively. A possible reason is that MCMN needs to meta-adapt with the support set. Yet a support set only has $N$-way $K$-shot samples, while NOTA exists in the query set. This tends to cause overfitting on the known instances but fails to detect NOTA. This further demonstrates the effectiveness and robustness of the proposed D-Proto model toward various NOTA rates.

Furthermore, we evaluate the performance of the ChatGPT[2] dialog system in the NOTA challenge. As demonstrated in Table 2, ChatGPT's performance on the NOTA task is far below other models. Therefore, we conclude that the model targeting the NOTA challenge in few-shot relation classification is still necessary. More details can be found in Appendix A.9.

### 3.4.2 NOTA Score Analysis

We further investigate the scores computed by D-Proto, which are depicted through box line diagrams in Figure 5. We first observe that the base scores alone are able to distinguish between known instances and NOTA ones, but the numerical values for the two classes are close. Specifically, the base scores of NOTA instances are generally greater than 0.4, while those of non-NOTA samples[3] are

around 0.1. This leads to difficulty in choosing a good threshold since the range of proper thresholds is small. Thus, base scores alone may not be sufficient to distinguish between the two classes accurately.

Secondly, without adding prototypes and degenerative prototypes, we have the naive local outlier factor (LOF). It can be seen that both LOF and PLOF show numerical overlap for both non-NOTA and NOTA instances. However, in LOF, the scores for non-NOTA instances are centered around 0.9, while the scores for NOTA instances range from 1.0 to 1.2. The available range of threshold is limited. Contrarily, in PLOF, the scores for non-NOTA samples are in the range of 2 to 3, while the scores for NOTA instances are significantly larger than 4.

Finally, combining PLOF and base NOTA score forms our final NOTA score. It is found that the NOTA score can make two types of instances have a clear difference. The gap between medians becomes more obvious, and the NOTA score for the NOTA instance becomes larger. These all contribute to the detection of NOTA instances, showing the effectiveness of our proposed NOTA score.

### 3.4.3 Ablation Study

To explore the impact of individual training objectives, we perform an ablation study. The results are reported in Table 3. It can be seen that removing $\mathcal{L}_{nota}$ causes significant decreases in all evaluation metrics. This verifies that NOTA loss effectively

---

[2]https://chat.openai.com
[3]In the context of the section, non-NOTA samples can be considered equivalent to known ones.

| Methods | 15%NOTA | | 30%NOTA | | 50%NOTA | | Average | |
|---|---|---|---|---|---|---|---|---|
| | Acc | F1 | Acc | F1 | Acc | F1 | Acc | F1 |
| D-Proto w/o $\mathcal{L}_{nota}$ | 84.95 | 78.73 | 84.04 | 78.27 | 82.72 | 75.41 | 83.90 | 77.47 |
| D-Proto w/o $Score_{base}$ | 78.44 | 70.95 | 78.23 | 71.39 | 77.52 | 68.99 | 78.06 | 70.44 |
| D-Proto w/o $\mathcal{P}$ | 84.24 | 78.00 | 83.93 | 78.27 | 83.48 | 76.13 | 83.88 | 77.47 |
| D-Proto w/o $\mathcal{DP}$ | 85.40 | 79.20 | 83.39 | 77.55 | 80.32 | 73.01 | 83.03 | 76.58 |
| D-Proto w/o $\{\mathcal{P} \cup \mathcal{DP}\}$ | 83.27 | 76.83 | 83.04 | 77.10 | 82.37 | 74.73 | 82.89 | 76.22 |
| D-Proto w. $20\%\mathcal{DP}_{K-2}$ | 84.04 | 77.76 | 83.80 | 78.10 | 83.46 | 76.10 | 83.77 | 77.32 |
| D-Proto w. $40\%\mathcal{DP}_{K-2}$ | 83.60 | 77.29 | 83.49 | 77.47 | 83.75 | 76.36 | 83.61 | 77.04 |
| D-Proto w. $60\%\mathcal{DP}_{K-2}$ | 83.35 | 76.99 | 83.63 | 77.88 | 83.88 | 76.49 | 83.62 | 77.12 |
| D-Proto w. $80\%\mathcal{DP}_{K-2}$ | 83.19 | 76.81 | 83.58 | 77.83 | **84.01** | **76.62** | 83.59 | 77.08 |
| D-Proto w. $100\%\mathcal{DP}_{K-2}$ | 83.70 | 77.37 | 83.74 | 78.02 | 83.72 | 76.37 | 83.72 | 77.25 |
| D-Proto(Ours) | **85.76** | **79.79** | **84.80** | **79.27** | 83.30 | 76.32 | **84.62** | **78.46** |

Table 3: Ablation study results and degenerative prototypes study (%) of D-Proto on FewRel benchmark.

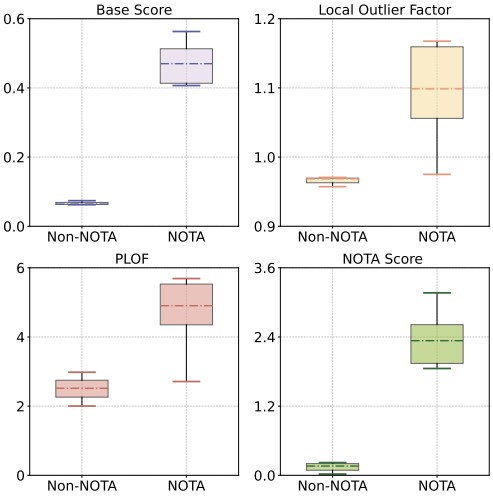

Figure 5: Comparison of all scores presented in this paper under 50% NOTA rate on the FewRel dataset. The dash-dotted line indicates the median of each score.

makes NOTA isolated, helping to recognize relations with densities.

Then removing the Base Score significantly impacts the model's performance. The Base Score is essential as it serves as the basis of PLOF. It can also be observed that the removal of either the prototype set $\mathcal{P}$ or the degenerative prototype set $\mathcal{DP}$ results in a significant decline in the evaluation metrics across different NOTA rates. Additionally, the removal of both $\mathcal{P}$ and $\mathcal{DP}$ results in an even more significant drop in the metrics. These findings suggest that both the prototype set and the degenerative prototype set are important components of the PLOF method and contribute significantly to its performance.

### 3.4.4 Degenerative Prototypes Analysis

We have a research question: *does adding more degenerative prototypes also increase the performance of the model?* To answer this question, we design the $K-2$ degenerative prototype set $\mathcal{DP}_{K-2}$. $\mathcal{DP}_{K-2}$ consists of all $K-2$ degenerative prototypes, where each one is computed by averaging the embeddings of the $K-2$ support instances belonging to its relation $r^i$ after excluding two support instances. It is worth noting that D-Proto only adds the degenerative prototypes $\mathcal{DP}_{K-1}$. Will including $\mathcal{DP}_{K-2}$ bring further gains?

In this experiment, we add various proportions of $\mathcal{DP}_{K-2}$ into the support set. The results are displayed in the second part of Table 3. Our expectation is that $\mathcal{DP}_{K-2}$ will increase the density of known samples and thus improve the performance. However, the results of this experiment show that while the performance of the D-Proto method does improve when $\mathcal{DP}_{K-2}$ is added under a NOTA rate of 50%, the performance decreases for NOTA rates of 15% and 30%. This suggests that while the use of $\mathcal{DP}_{K-2}$ can indeed improve the detection of NOTA instances at higher densities, it also leads to the false detection of known instances as NOTA, particularly at lower NOTA rates.

### 3.4.5 Visualization

To further evaluate the representations learned by different models, we visualize the embeddings using the t-SNE algorithm (Van der Maaten and Hinton, 2008) in Figure 6. Under both the 15% and 30% NOTA rates, compared with O-Proto and MNAV, our method can make each relation cluster more densely. For instance, in Figure 6 (a), the relation 1 is scattered around in O-Proto and MNAV. However, in our method, this class is well aggregated. This shows that our method can learn more separable representations for known instances.

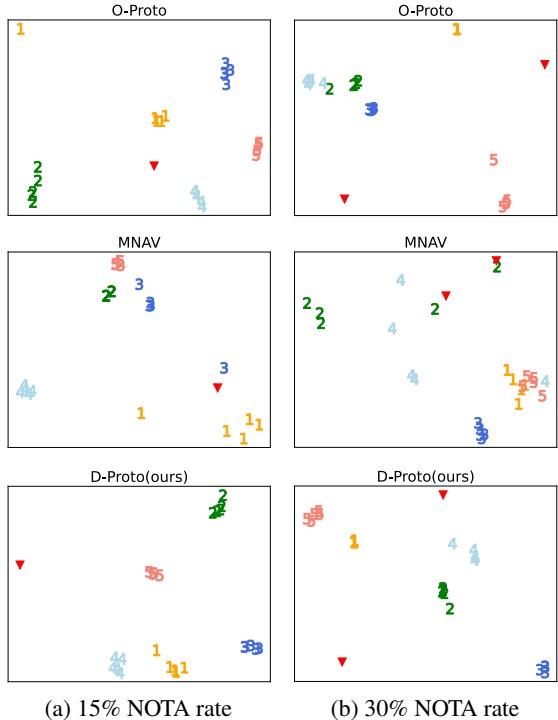

(a) 15% NOTA rate     (b) 30% NOTA rate

Figure 6: Embedding distributions of different models in the same episode on the FewRel dataset. NOTA rates are 15% and 30%, respectively. The numbers 1 to 5 represent the relation. The red inverted triangles are NOTA instances.

Moreover, in Figure 6 (a), we see that the NOTA instance is far away from known samples in our method. This observation is visible in the plot (b). This shows the validity of the proposed NOTA loss $\mathcal{L}_{nota}$, making NOTA instances more isolated. In MNAV, the two NOTA instances are close to a known sample. One possible reason is that MNAV makes multiple NOTA instances approach learnable vectors. Different classes may aggregate with each other. This tends to cause negative effects on distinguishing samples, as embeddings are not well separated. In summary, our method learns the best representations. With the well-clustered known samples and the isolated NOTA samples, D-Proto can both recognize them through the density-aware NOTA detection module accurately.

## 4 Related Work

Previous works (Zelenko et al., 2003; Mooney and Bunescu, 2005; Zeng et al., 2014; Gormley et al., 2015; Zhou et al., 2016; Kumar, 2017; Zhang et al., 2018) for relation classification usually train models on a labeled dataset with a fixed number of classes, which cannot deal with unseen relations.

Thus few-shot scenario has drawn much attention recently. By training on multiple episodes, models can learn transferrable knowledge for unseen testing episodes (Hu et al., 2021). Although many efforts (Hendrickx et al., 2010; Zhang et al., 2017; Obamuyide and Vlachos, 2019; Gao et al., 2019a; Baldini Soares et al., 2019; Ye and Ling, 2019; Dong et al., 2020) have been devoted to the few-shot relation classification. However, a real-world issue that has been proposed is the NOTA challenge (Gao et al., 2019b). This issue breaks the assumption of the traditional $N$-way $K$-shot setting. Therefore, detecting NOTA instances during inference needs to be explored.

Promising works have been proposed to detect NOTA instances in few-shot relation classification. One approach that has been widely adopted is the $N + 1$ classification. This approach has been proposed by several works. One work is to use a sentence-pair model for classification, as proposed by Gao et al. (2019b) in their work on few-shot learning. Another work, Sabo et al. (2021) in their work on revisiting NOTA challenge, is to compute the distance between learnable vectors. The third one, proposed by Liu et al. (2022), adopts the pre-training fine-tuning paradigm and selects the correct relation from a set of options that includes NOTA. The above three methods, by considering NOTA instances as an additional class, are able to learn the boundary between the known relations and the NOTA instances, allowing the model to learn a more comprehensive representation of the data.

## 5 Conclusion

In this paper, we propose D-Proto, a density-aware prototypical network for few-shot relation classification. We focus on the overall instance distribution and design special training objectives for distinct samples. For the known samples, intra-class becomes well-clustered and inter-class is well-separated. The NOTA ones are isolated as outliers. We further propose a NOTA detection module, which considers the local density and distance property, to distinguish both types of instances. And we use prototypes and degenerative prototypes to enhance the density of known instances to better identify NOTA. Experiments on a popular dataset demonstrate that D-Proto outperforms strong baselines. We also provide score analysis and visualization to verify the effectiveness of our method.

## Limitations

In this work, we study the few-shot relation classification task, which contains NOTA instances. The proposed D-Proto method achieves some gains on this problem. Yet our work still has the following two limitations.

The first limitation is that NOTA instances are detected with a threshold $\tau$. This threshold relies on human experiences or k-fold validation. However, using a threshold in the binary way methods, including O-Proto and ours, is not a big problem. Our viewpoint is that $N + 1$ classification can efficiently accomplish the NOTA detection task by treating the NOTA as an additional class to classify with the known $N$ classes. But the $N + 1$ classification method ignores the distribution of the data. Therefore, despite that we need a threshold, the advantages of our method are also obvious.

The second limitation is that our experiment is not absolutely fair. The reason is that BERT-Pair (Gao et al., 2019b) requires high GPU memory. In the fixed sampling method of the episode, we used two NVIDIA RTX A6000 with 48G memory each to train BERT-Pair. The other baselines are trained with an NVIDIA TESLA V100-32G. Except for this difference, all other settings are the same.

## Acknowledgment

We sincerely thank all the anonymous reviewers for providing valuable feedback. This work is supported by the National Natural Science Foundation of China (Grant No. 62302245), the Ministry of Education of the People's Republic of China Humanities and Social Sciences Youth Foundation (Grant No. 23YJCZH240), and the youth program of National Science Fund of Tianjin, China (Grant No. 22JCQNJC01340).

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

| | Random Sampling | | | | | | | |
|---|---|---|---|---|---|---|---|---|
| Methods | 15%NOTA | | 30%NOTA | | 50%NOTA | | Average | |
| | Acc | F1 | Acc | F1 | Acc | F1 | Acc | F1 |
| O-Proto | 80.02±0.65 | 72.84±0.88 | 80.55±0.97 | 73.73±1.18 | 81.16±1.04 | 72.41±1.28 | 80.58 | 72.99 |
| BERT-Pair | 80.77±2.35 | 73.99±2.67 | 81.53±1.60 | 75.06±1.92 | 83.24±0.85 | 75.14±1.38 | 81.85 | 74.73 |
| MNAV | 85.02±0.39 | 78.81±0.47 | 83.67±0.66 | 77.78±0.90 | 81.59±1.02 | 73.94±1.18 | 83.42 | 76.84 |
| MCMN | **88.24±1.83** | **83.54±2.45** | 82.50±3.40 | 76.39±4.34 | 74.88±5.83 | 67.49±6.64 | 81.87 | 75.80 |
| ChatGPT | 64.57±0.31 | 56.63±0.41 | 63.30±0.57 | 55.81±0.40 | 61.66±0.76 | 54.16±0.45 | 63.18 | 55.54 |
| D-Proto | 85.76±0.72 | 79.79±0.96 | **84.80±0.92** | **79.27±1.12** | **83.30±1.29** | **76.32±1.48** | **84.62** | **78.46** |
| | Fixed Sampling | | | | | | | |
| Methods | 15%NOTA | | 30%NOTA | | 50%NOTA | | Average | |
| | Acc | F1 | Acc | F1 | Acc | F1 | Acc | F1 |
| O-Proto | 81.38±1.26 | 80.45±1.29 | 81.42±1.11 | 81.04±1.19 | 81.38±0.96 | 79.39±1.06 | 81.38 | 80.29 |
| BERT-Pair | 82.83±0.62 | 81.98±0.62 | 83.76±0.46 | 83.00±0.54 | **85.49±0.29** | 82.62±0.45 | 84.02 | 82.53 |
| MNAV | 85.05±0.97 | 84.22±1.03 | 84.36±1.34 | 84.45±1.15 | 82.91±2.61 | 82.15±1.85 | 84.10 | 83.60 |
| MCMN | **87.56±1.93** | 84.04±2.52 | 83.16±3.37 | 83.40±3.08 | 75.07±5.79 | 79.14±3.93 | 81.93 | 82.19 |
| ChatGPT | 63.59±0.50 | 56.26±0.27 | 61.83±0.41 | 54.46±0.32 | 58.41±0.51 | 51.14±0.43 | 61.28 | 53.95 |
| D-Proto | 85.37±0.61 | **84.67±0.61** | **84.87±0.71** | **84.90±0.66** | 83.05±1.00 | **82.66±1.10** | **84.43** | **84.08** |

Table 4: Evaluation results of baseline methods and the proposed D-Proto, in terms of accuracy (%) and F1 (%), on the FewRel dataset. This table includes the standard deviation of baseline methods and D-Proto.

## A  Reproducibility

### A.1  Implementation Details

Following Gao et al. (2019b), we exploit the same hyper-parameters. For a fair comparison, all baselines and our work use the same encoder. We choose **BERT-base-uncased** (Devlin et al., 2019) as the encoder. It is initialized by the pre-trained parameters and optimized during training. For the FewRel dataset, we set the batch size to 2, which means we feed two episodes into the model per batch. Each model is trained for 30 epochs with 1000 batches per epoch and tested with 10000 batches in the testing. An early stopping strategy is adopted, indicating the model will stop training when the performance on the validation set does not improve for 6 epochs. The best model on the validation set is saved for evaluation.

In Eq. (1), we set the hyper-parameters $\alpha$ to 1e-5 on the FewRel. For $k$ of the local outlier factor in Eq. (7), we set it to be $K$. The reason is that in the $N$-way $K$-shot setting, a support instance has other $(K - 1)$ neighbors with the same relation. And each relation has at least one prototype. This is consistent with the definition of $k$-neighborhood. All the reported results using the FewRel dataset are the average of five runs with fixed seeds [5, 10, 15, 20, 25].

After the hyperparameter search, the threshold $\tau$ is set to 0.9 in the experimental setting of FewRel.

For MCMN, the sampling method in the pre-

| Model | Number of parameters |
|---|---|
| O-Proto | 109482240 |
| MNAV | 109497600 |
| BERT-Pair | 109483778 |
| D-Proto | 109482240 |

Table 5: The number of hyperparameters for each model.

training follows Liu et al. (2022), while the sampling method in the testing is fixed sampling or random sampling.

### A.2  Computing Infrastructure

Since BERT-Pair (Gao et al., 2019b) requires high GPU memory in the fixed sampling of 5-way 5-shot setting, we use 2 NVIDIA A6000s with 48G memory each to train the model. The rest of the baselines and our model are trained on a NVIDIA TESLA V100 with 32G of GPU memory.

### A.3  Number of Parameters per Model

Table 5 shows the number of hyperparameters for each model.

### A.4  Evaluation Metrics

We use two primary evaluation metrics, accuracy and macro F1. The basic implementation of accu-

| Method | Number of instances |
|---|---|
| D-Proto w/o $\mathcal{P}$ | 53 |
| D-Proto w/o $\mathcal{DP}$ | 33 |
| D-Proto w/o $\{\mathcal{P} \cup \mathcal{DP}\}$ | 28 |
| D-Proto w. 20%$\mathcal{DP}_{K-2}$ | 68 |
| D-Proto w. 40%$\mathcal{DP}_{K-2}$ | 78 |
| D-Proto w. 60%$\mathcal{DP}_{K-2}$ | 88 |
| D-Proto w. 80%$\mathcal{DP}_{K-2}$ | 98 |
| D-Proto w. 100%$\mathcal{DP}_{K-2}$ | 108 |
| D-Proto(Ours) | 58 |

Table 6: The number of instances for each method in an episode. The experiment setting is 5-way 5-shot. The number of query instances is 3.

racy is as follows:

$$Accuracy = \frac{TP + TN}{TP + FP + TN + FN}$$

where $TP$ are true positives, $FP$ are false positives, $TN$ are true negatives, and $FN$ are false negatives. The definition of F1 is as follows:

$$P = \frac{TP}{TP + FP}$$
$$R = \frac{TP}{TP + FN}$$
$$F1 = \frac{2 \times P \times R}{P + R}$$

Macro F1 calculates F1 for each class and finds their unweighted mean.

## A.5 Hyperparameters

Following Gao et al. (2019b), the hyperparameters not mentioned in Section A.1 are as follows:

- Learning rate: 2e-5.

- Weight decay: 0.01.

- Warmup steps: 300.

- Gradient accumulation: 1.

- Max length: 128.

- Hidden size: 768.

- Test batch: 10000.

## A.6 Links to Dataset

FewRel (Han et al., 2018): https://thunlp.github.io/2/fewrel2_nota.html.

## A.7 Links to Toolkit

- BERT: https://huggingface.co/transformers/v4.5.1/model_doc/bert.html#bertmodel

- BERT Tokenizer: https://huggingface.co/transformers/v4.5.1/model_doc/bert.html#berttokenizer

- Local Outlier Factor(LOF): https://scikit-learn.org/stable/modules/generated/sklearn.neighbors.LocalOutlierFactor.html

- Macro F1: https://scikit-learn.org/stable/modules/generated/sklearn.metrics.f1_score.html

## A.8 Number of Instances per Method

Table 6 shows the number of instances included in an episode for each method in ablation studying. The episode sampling method is random sampling.

## A.9 Prompt of ChatGPT

We demonstrate the prompt for using ChatGPT directly in the NOTA task, as follows:

> **Question**: *The purpose of the relation classification task is to identify the relation between two entities (one is the head entity and the other is the tail entity) in a given sentence. For example, the following sentence "California is a state in the United States". Through the derivation of the whole sentence, the relation between the head entity "California" and the tail entity the "United States" is "contains". While the few-shot relation classification task consists of a support set and a query set. The support set generally contains $N$ relations, and each relation contains $K$ instances. This setup is called $N$-way $K$-shot. We need to predict the relation between two entities in each sentence given in the query set based on the support set. Now I need you to complete a few-shot relation classification task, which is set to 5-way 5-shot. The five categories in the support set are numbered from 0 to 4, respectively. Five sentences are given after*

*each category. The head entity of each sentence is surrounded by two special characters [head] and [/head], and the tail entity is surrounded by two special characters [tail] and [/tail]. You need to predict the relationship between two entities for each sentence in the query set from the support set. That is, if the relation of a query instance is similar to that of all instances of category 0 in the support set, then the category predicted by the query instance is category 0. Notice! If the relation between the two entities of a sentence in the query set cannot be referenced in the support set, the relation between the two entities of the sentence is called NOTA (None-of-the-above). When the NOTA relation is encountered, the category number of the predicted result is 5. Warning! You just need to return the class number for each instance predicted in the query set. For example, the relation of the first instance in the query set is relation 0, and the second instance is relation 5 (NOTA). Then your reply is: [0, 5]. If you get it, then I'll give you support sets and query sets next.*

**ChatGPT**: *Yes, I understand the task. Please provide me with the support set and query set.*

**Question**: *The following is the support set (5-way 5-shot. Instances in each category are separated by a special character [sep].):*
*......*
*The following is the query set (The query set contains one instance.):*
*......*