# OpenReview forum: "Density-Aware Prototypical Network for Few-Shot Relation Classification"
_EMNLP/2023/Conference — EMNLP 2023 Findings_

### Official Review · Reviewer_sq7x · 2023-08-03

**Typos Grammar Style And Presentation Improvements:** No
**Soundness:** 4

**Excitement:**

4: Strong: This paper deepens the understanding of some phenomenon or lowers the barriers to an existing research direction.

**Missing References:**

No

**Paper Topic And Main Contributions:**

This work investigates the subproblem none-of-the-above (NOTA) under the field of few-shot. The authors notice that previous approaches have not focused on the discrepancy between the data distribution of NOTA and the known categories in small samples, and based on this they propose density-aware prototypical network (D-Proto) to leverage the distributional difference, claiming that is the first to model the overall instance distribution in an episode. They also propose NOTA loss to balance the distributional differences between the two heterogeneous data, then further use a prototype enhanced local outlier factor (PLOF) and a base score to fully leverage the density properties. Extensive experimental results demonstrated the effectiveness and robustness of the proposed method.

**Questions For The Authors:**

Question A: Could you provide more informative arguments or illustration to demonstrate the distribution imbalance of the chosen dataset distribution or any other dataset?

**Reasons To Accept:**

For the NOTA problem, D-Proto innovatively starts from the data distribution and rigorously defines a series of new loss functions and methods to answer the question "what instances distribution is ideal". It progressively combines known sample separation, NOTA sample isolation, and NOTA detection to leverage the density properties. Detailed experiments and graphs illustrate the effectiveness of each of the proposed approaches on both the Random and Fixed sampling methods, where the F1 score and Accuracy outperform the previous studies.

**Reasons To Reject:**

The authors only tested the proposed method on the FewRel dataset. Considering the specificity of the NOTA detection task, more experiments on other datasets should be included to validate the effectiveness of D-Proto. In addition, the article lacks a comparison of the distributional differences of the original data, although a comparison between the proposed D-Proto and the two previous amplifications is provided in the visualization section, the analysis of the original dataset is even more important since it is the main source of this research’s intuition, which should have been given more justification and explanation.

**Reproducibility:**

4: Could mostly reproduce the results, but there may be some variation because of sample variance or minor variations in their interpretation of the protocol or method.

**Reviewer Confidence:**

4: Quite sure. I tried to check the important points carefully. It's unlikely, though conceivable, that I missed something that should affect my ratings.

---

> ### Author Rebuttal · Authors · 2023-08-27
>
> Thanks for your valuable comments and suggestions.
>
> __Q1:__ The authors only tested the proposed method on the FewRel dataset. Considering the specificity of the NOTA detection task, more experiments on other datasets should be included to validate the effectiveness of D-Proto.
>
> __A1:__ We appreciate your kind suggestions. Although our experiments only used the FewRel dataset, which is a publicly available dataset, it is currently the highest-quality and most balanced dataset for the FSRC task. Additionally, our experiments employed multiple data sampling methods and different NOTA rates. The extensive ablation study also serves as evidence for the effectiveness of our approach. We will conduct more experiments by using more FSRC datasets in the camera-ready version.
>
> __Q2:__ In addition, the article lacks a comparison of the distributional differences of the original data, although a comparison between the proposed D-Proto and the two previous amplifications is provided in the visualization section, the analysis of the original dataset is even more important since it is the main source of this research’s intuition, which should have been given more justification and explanation.
>
> __A2:__ Thank you for pointing out the issue that we overlooked. The FewRel dataset proposed by FewRel 1.0[1] contains $700$ instances for each category, so its distribution is balanced. When we mention imbalanced distribution, we are referring to the phenomenon of imbalance within each sampled episode. For the specific cases, please refer to answer __A3__.
>
> __Q3:__ Could you provide more informative arguments or illustration to demonstrate the distribution imbalance of the chosen dataset distribution or any other dataset?
>
> __A3:__ We appreciate your kind suggestions. Below, we provide an example sampled from the FewRel dataset. The query set includes $5$ instances of non-NOTA (not given) and $5$ instances of NOTA, each belonging to $5$ different categories. If we don't consider the distribution and cluster these NOTA instances of different categories together (like in MNAV), it will inevitably lead to an imbalance in the instances of this episode. In the camera-ready version, we will provide further analysis to illustrate this imbalance, such as a case study.
>
> Support set:
>
> {
>
> “Record label”: {……}
>
> “Operator”: {……}
>
> “Moment”: {……}
>
> “Followed by”: {……}
>
> “Country”: {……}
>
> }
>
> Query set:
>
> {
>
> ……
>
> 1. [head] Miss Navajo [\head] Nation is a pageant that has been held annually on the Navajo Nation, [tail] United States [\tail], since 1952.
> 2. The [tail] Wildcats [\tail] moved off campus from Memorial Coliseum to [head] Rupp Arena [\head] in the downtown metroplex.
> 3. Actress Sally Hawkins was asked to play "[head] Persuasion [\head] 's" protagonist [tail] Anne Elliot [\tail].
> 4. The songs were re-mastered to vinyl by Robert Hadley and former lead singer of [head] Journey [\head], [tail] Steve Perry [\tail].
> 5. [head] William Trevor [\head], "[tail] The Story of Lucy Gault [\tail]" , 2002, page 14.
>
> }
>
> References:
>
> [1] Han X, Zhu H, Yu P, et al. FewRel: A Large-Scale Supervised Few-Shot Relation Classification Dataset with State-of-the-Art Evaluation[J].

---

### Official Review · Reviewer_sQUS · 2023-08-05

**Soundness:** 3

**Excitement:**

3: Ambivalent: It has merits (e.g., it reports state-of-the-art results, the idea is nice), but there are key weaknesses (e.g., it describes incremental work), and it can significantly benefit from another round of revision. However, I won't object to accepting it if my co-reviewers champion it.

**Paper Topic And Main Contributions:**

This paper studies the important problem of few-shot relation classification where there exist a substantial number of none-of-the-above (NOTA) samples, which do not belong to any classes. The authors propose to identify NOTA samples as outliers that differ from the distribution of known samples. By designing a unique training objective that separate known samples and NOTA samples, the authors manage to accurately detect the existence of NOTA samples. The authors also conduct extensive experiments to verify the effectiveness of their framework.

**Questions For The Authors:**

Have the authors considered testing their framework under common scenarios (i.e., without NOTA samples)? The datasets included in the experiments are also insufficient. Do the authors consider adding more datasets for comparison?

**Reasons To Accept:**

1. This work studies the important problem of few-shot relation classification under the scenario with existence of NOTA samples. The experiments also validate the effectiveness of the framework.

2. The authors design specific strategies that are inspiring for future work on few-shot relation classification. The authors also investigate the challenges in few-shot relation classification, which can be crucial for future research.

3. The paper is well-written and easy to follow.

**Reasons To Reject:**

1. The authors mainly focus on the scenario where there exist a substantial number of NOTA samples. However, such a scenario may not be very common in real world. Specifically, in the experiments, the authors show results with more than 15% NOTA samples, which can be a lot in realistic scenarios. More importantly, the authors do not provide justification such as references of such existence of a high ratio of NOTA samples.

2. The experimental part is not solid. Specifically, the compared methods are not enough (only four few-shot relation classification baselines compared with two before 2020). Moreover, these methods are not specifically proposed for scenarios with such a high ratio of NOTA samples, which will reduce the fairness of the experiments.

3. Most importantly, although the authors claim that they focus on scenarios with NOTA samples, they do not compare the results of their framework and other baselines regarding the scenario without NOTA samples, which should be more common in realistic scenarios. The lack of such comparison harms the soundness of experiments

**Reproducibility:**

3: Could reproduce the results with some difficulty. The settings of parameters are underspecified or subjectively determined; the training/evaluation data are not widely available.

**Reviewer Confidence:**

4: Quite sure. I tried to check the important points carefully. It's unlikely, though conceivable, that I missed something that should affect my ratings.

---

> ### Author Rebuttal · Authors · 2023-08-27
>
> Thanks for your valuable comments and suggestions.
>
> __Q1:__ The authors mainly focus on the scenario where there exist a substantial number of NOTA samples. However, such a scenario may not be very common in real world. Specifically, in the experiments, the authors show results with more than 15% NOTA samples, which can be a lot in realistic scenarios. More importantly, the authors do not provide justification such as references of such existence of a high ratio of NOTA samples.
>
> __A1:__ Thanks for pointing this out. FewRel 2.0[2] first introduced the NOTA problem and provided standard experimental settings with NOTA rates of 15%, 30%, and 50% respectively. Our experiments followed these standard settings. We will conduct further experiments to explore the effectiveness of our proposed method at lower NOTA rates.
>
> __Q2:__ The experimental part is not solid. Specifically, the compared methods are not enough (only four few-shot relation classification baselines compared with two before 2020). Moreover, these methods are not specifically proposed for scenarios with such a high ratio of NOTA samples, which will reduce the fairness of the experiments.
>
> __A2:__ As mentioned in answer __A1__, the baseline we selected is also based on the standard settings of FewRel 2.0[2]. We will provide further results in the camera-ready version.
>
> __Q3:__ Most importantly, although the authors claim that they focus on scenarios with NOTA samples, they do not compare the results of their framework and other baselines regarding the scenario without NOTA samples, which should be more common in realistic scenarios. The lack of such comparison harms the soundness of experiments
>
> __A3:__ This is a good question. The absence of NOTA instances is a classic setting in the FSRC task proposed by FewRel 1.0[1]. The new NOTA problem introduced by FewRel 2.0 [2] is currently receiving the most attention, as the NOTA task aligns more with real-world situations and presents greater difficulty.
>
> __Q4:__ Have the authors considered testing their framework under common scenarios (i.e., without NOTA samples)? The datasets included in the experiments are also insufficient. Do the authors consider adding more datasets for comparison?
>
> __A4:__ We appreciate your kind suggestions. As mentioned in answer __A3__, we will further investigate the performance of D-Proto on traditional FSRC tasks and provide more experimental results in the camera-ready version.
>
> References:
>
> [1] Han X, Zhu H, Yu P, et al. FewRel: A Large-Scale Supervised Few-Shot Relation Classification Dataset with State-of-the-Art Evaluation[J].
>
> [2] Gao T, Han X, Zhu H, et al. FewRel 2.0: Towards More Challenging Few-Shot Relation Classification[J].

---

### Official Review · Reviewer_vjzP · 2023-08-15

**Soundness:** 4

**Excitement:**

4: Strong: This paper deepens the understanding of some phenomenon or lowers the barriers to an existing research direction.

**Paper Topic And Main Contributions:**

The D-Proto network addresses the problem of none-of-the-above (NOTA) relation classification by treating various instances distinctly. Specifically, the authors design unique training objectives to separate known instances and isolate NOTA instances, respectively. This produces an ideal instance distribution, where known instances are dense yet NOTAs have a small density. Moreover, the authors propose a NOTA detection module to further enlarge the density of known samples, and discriminate NOTA and known samples accurately. The experimental results of the proposed method show that D-Proto outperforms strong baselines and achieves the best average results on both fixed and random sampling strategies.

**Reasons To Accept:**

1. Novelty: The paper proposes a novel density-aware prototypical network (D-Proto) for few-shot relation classification, which addresses the challenging problem of none-of-the-above (NOTA) relation classification, which is a fundamental task in natural language processing.

2. Effectiveness: The experimental results demonstrate that D-Proto outperforms strong baselines with robustness towards various NOTA rates. This suggests that D-Proto can be a promising solution for few-shot relation classification tasks when it comes to NOTA problem.

3. Clarity: The paper is well-written and organized, with clear explanations of the proposed method and experimental results. The authors also provide score analysis and visualization to verify the effectiveness of their method.

4. Reproducibility: The authors provide a clear reproducibility for further replication and plan to make the code public after the paper is accepted, which can facilitate further research and development in this area.

**Reasons To Reject:**

The weaknesses of this paper are as follows:

1. Limited evaluation: The proposed method is evaluated on only one dataset, which may limit the generalizability of the results. Further evaluation on other datasets is needed to validate the effectiveness of the proposed method.

2. Although the paper discusses the limitations of the proposed method, the discussion is relatively brief and lacks a detailed analysis of the limitations. A more in-depth discussion of the error analysis can help readers better understand the potential challenges and future directions of the research.

3. In the experimental section, it would be clearer to separately highlight the improvements in NOTA cases attributed to the proposed D-proto. A more in-depth analysis of how D-proto addresses NOTA cases would offer a clearer explanation of the model's effectiveness.

**Reproducibility:**

4: Could mostly reproduce the results, but there may be some variation because of sample variance or minor variations in their interpretation of the protocol or method.

**Reviewer Confidence:**

3: Pretty sure, but there's a chance I missed something. Although I have a good feel for this area in general, I did not carefully check the paper's details, e.g., the math, experimental design, or novelty.

---

> ### Author Rebuttal · Authors · 2023-08-27
>
> Thanks for your valuable comments and suggestions.
>
> __Q1:__ Limited evaluation: The proposed method is evaluated on only one dataset, which may limit the generalizability of the results. Further evaluation on other datasets is needed to validate the effectiveness of the proposed method.
>
> __A1:__ We appreciate your kind suggestions. Although our experiments only used the FewRel dataset, which is a publicly available dataset, it is currently the highest-quality and most balanced dataset for the FSRC task. Additionally, our experiments employed multiple data sampling methods and different NOTA rates. The extensive ablation study also serves as evidence for the effectiveness of our approach. We will conduct more experiments by using more FSRC datasets in the camera-ready version.
>
> __Q2:__ Although the paper discusses the limitations of the proposed method, the discussion is relatively brief and lacks a detailed analysis of the limitations. A more in-depth discussion of the error analysis can help readers better understand the potential challenges and future directions of the research.
>
> __A2:__ Thank you for pointing out this issue. In the camera-ready version, we will introduce more analysis, such as conducting further error analysis. Below, we provide an example of a misclassified relation by the D-Proto model sampled from the FewRel dataset. The true category of query instance $1$ is “Country of origin”, but due to the semantic similarity between this instance and the relation “Country” in the support set, D-Proto failed to correctly detect it as the NOTA category.
>
> Support set:
>
> {
>
> “Record label”: {……}
>
> “Operator”: {……}
>
> “Moment”: {……}
>
> “Followed by”: {……}
>
> “Country”: {
> 1. [tail] Belgium [\tail]'s highest point is the [head] Signal de Botrange [\head] at 694 meters above the sea level.
> 2. [head] Syria [\head] was an early Roman province, annexed to the [tail] Roman Republic [\tail] in 64 BC by Pompey in the Third Mithridatic War, following the defeat of Armenian King Tigranes the Great.
> 3. The [head] Dniester [\head] rises in [tail] Ukraine [\tail], near the city of Drohobych, close to the border with Poland, and flows toward the Black Sea.
> 4. The Ulster House Hotel, formerly the Wellington Hotel, is located on Main Street in Pine Hill, New York, United States.
> 5. [head] Pico Turquino [\head] is the highest summit of the island and [tail] Republic of Cuba [\tail].
>
> }
>
> }
>
> Query set:
>
> {
>
> 1.	[head] Miss Navajo [\head] Nation is a pageant that has been held annually on the Navajo Nation, [tail] United States [\tail], since 1952.
>
> ……
>
> }
>
> __Q3:__ In the experimental section, it would be clearer to separately highlight the improvements in NOTA cases attributed to the proposed D-proto. A more in-depth analysis of how D-proto addresses NOTA cases would offer a clearer explanation of the model's effectiveness.
>
> __A3:__ Thanks for your suggestion. In Figure 6, we present specific examples by visualizing their representations. This figure demonstrates the improvements brought by our proposed D-Proto compared to other prototype-based baselines in terms of density and the distribution of NOTA. In the camera-ready version, we will provide further analysis with more cases.

---

### Meta-Review · Area_Chair_rbvE · 2023-09-19

**Recommendation:** 3

**Metareview:**

This paper aims to deal with few-shot relation extraction, especially to solve the problem of none-of-the-above (NOTA) relation classification. Specifically, this paper adopts an additional training objective to classify known and out-of-distribution instances, making the NOTA relations easier to identify. The structure of the article is well organized, and the presentation is relatively straightforward. The main problem of this paper is that the used datasets are limited, making the results not solid enough.

---

### Decision · Program_Chairs · 2023-10-07

**Decision:**

Accept-Findings

**Comment:**

This paper aims to deal with few-shot relation extraction, especially to solve the problem of none-of-the-above (NOTA) relation classification. Specifically, this paper adopts an additional training objective to classify known and out-of-distribution instances, making the NOTA relations easier to identify. The structure of the article is well organized, and the presentation is relatively straightforward. The main problem of this paper is that the used datasets are limited, making the results not solid enough.